

# The application of plant growth-promoting rhizobacteria in *Solanum lycopersicum* production in the agricultural system: a review

Afeez Adesina Adedayo[1], Olubukola Oluranti Babalola[1],
Claire Prigent-Combaret[2], Cristina Cruz[3], Marius Stefan[4], Funso Kutu[5] and Bernard R. Glick[6]

[1] Food Security and Safety Focus Area, Faculty of Natural and Agricultural Sciences, North-West University, Mmabatho, South Africa
[2] Microbial Ecology, University Lyon 1, The 'Rhizosphere' Team, Lyon, Villeurbanne, France
[3] Department of Plant Biology, Faculdade de Ciências, Universidade de Lisboa, Lisboa, Portugal
[4] Faculty of Biology, Universitatea Alexandru Ioan Cuza, Iasi, Romania
[5] Faculty of Agiculture and Natural Sciences, University of Mpumalanga, Mpumalanga, South Africa
[6] Department of Biology, University of Waterloo, Waterloo, Ontario, Canada

Corresponding author
Olubukola Oluranti Babalola,
olubukola.babalola@nwu.ac.za

## ABSTRACT

Food safety is a significant challenge worldwide, from plantation to cultivation, especially for perishable products such as tomatoes. New eco-friendly strategies are needed, and beneficial microorganisms might be a sustainable solution. This study demonstrates bacteria activity in the tomato plant rhizosphere. Further, it investigates the rhizobacteria's structure, function, and diversity in soil. Rhizobacteria that promote the growth and development of tomato plants are referred to as plant growth-promoting bacteria (PGPR). They form a series of associations with plants and other organisms in the soil through a mutualistic relationship where both parties benefit from living together. It implies the antagonistic activities of the rhizobacteria to deter pathogens from invading tomato plants through their roots. Some PGPR are regarded as biological control agents that hinder the development of spoilage organisms and can act as an alternative for agricultural chemicals that may be detrimental to the health of humans, animals, and some of the beneficial microbes in the rhizosphere soil. These bacteria also help tomato plants acquire essential nutrients like potassium (K), magnesium (Mg), phosphorus (P), and nitrogen (N). Some rhizobacteria may offer a solution to low tomato production and help tackle food insecurity and farming problems. In this review, an overview of soil-inhabiting rhizobacteria focused on improving the sustainable production of *Solanum lycopersicum*.

## INTRODUCTION

The health status of the microbial communities present in the soil environment depends on the soil's quality. The soil's health status promotes its agricultural sustainability (*Odelade & Babalola, 2019*). Various studies have demonstrated the effects of

microorganisms on tomatoes regarding their size and development, proper seed multiplication, nutrition, disease resistance, and seedling development (*De Coninck et al., 2021*; *Patil & Fauquet, 2021*). In addition, the soil factors affecting plant growth are as follows; dissolved oxygen concentration, nutrients, phytopathogens and parasites, water, and weed seed pools (*Patil & Fauquet, 2021*).

Microbiota is the community of bacteria, archaea, and fungi that inhabit a particular environment, especially in soil. They are also referred to as a collection of microorganisms living in or on the organism's tissue. Plants are well known as distinct organisms that carry microbiota (*Berg et al., 2020*). The plant-microbe interactions shown in Fig. 1 provide a better knowledge of mutual relationships involved in inter-kingdom networks of feeding on the substrate manufactured in the rhizosphere. The presence of PGPR facilitates the development of the size and health of tomatoes.

Various interactions occur between microbial species in the rhizosphere of healthy growing tomatoes (*Jain, Chakraborty & Das, 2020*). The rhizospheric community is susceptible to changes in nature. This state shows the features of the microbial populations present and reveals the biological balance between them (*Jansson & Hofmockel, 2020*). The interaction of the tomato plant with microorganisms takes place in a certain way that permits the coexistence of favorable species. This coexistence and relationships are regarded as the norm in nature. Some microorganisms introduced are often alive in the new region they colonized because of their interaction with other rhizospheric populations. Because of these interrelationships, introduced microbes are nonetheless rarely sustained in the new environment they occupy (*Odelade & Babalola, 2019*).

Plant roots with bacteria have a beneficial impact on the development and production of crop plants (*Fasusi, Cruz & Babalola, 2021*). The syndicate of bacterial species is the PGPR that lives in the soil found around the plant's root, influencing plant development and is profitable health-related. They are agricultural biological resources that induce the plant's growth and fruitfulness. They also motivate resistance in plants, *i.e.*, a wide range of vegetation of fruits, vegetables, and various forest trees, to different phytopathogens (*Zia et al., 2020*).

The existence of bacterial species in the rhizosphere soil has been applied as a biological signal to calculate the soil's quality and fertility. These bacteria are regarded as biofertilizer that causes no harm to the edaphic profile and ecological sustainability. They are regarded as PGPR, they are known to produce phytohormones and introduce specific functions in sustainable agriculture. Aside from the phytohormone they produced, a lot of crucial functions which include; fixing atmospheric nitrogen, and distributing essential food substances are among the various functions they carry out in the rhizosphere (*Fasusi, Cruz & Babalola, 2021*). They have improved the organic carbon content, the composition of water, soil acidity and alkalinity, and soil porosity (*Glick, 2020b*).

The sole earthy ecological niche where microorganisms inhabited, and their arrangement deviates from plant species, is known as the rhizosphere soil (*Agbaji, Nwaichi & Abu, 2021*). The most prevalent bacterial diversity associated with the root found in the rhizosphere soil is *Alphaproteobacteria, Betaproteobacteria, Gammaproteobacteria, Bacteroidetes* (*e.g., Rhizobia, Burkholderia, Pseudomonas*), and *Firmicutes* (*e.g., Bacillus*)

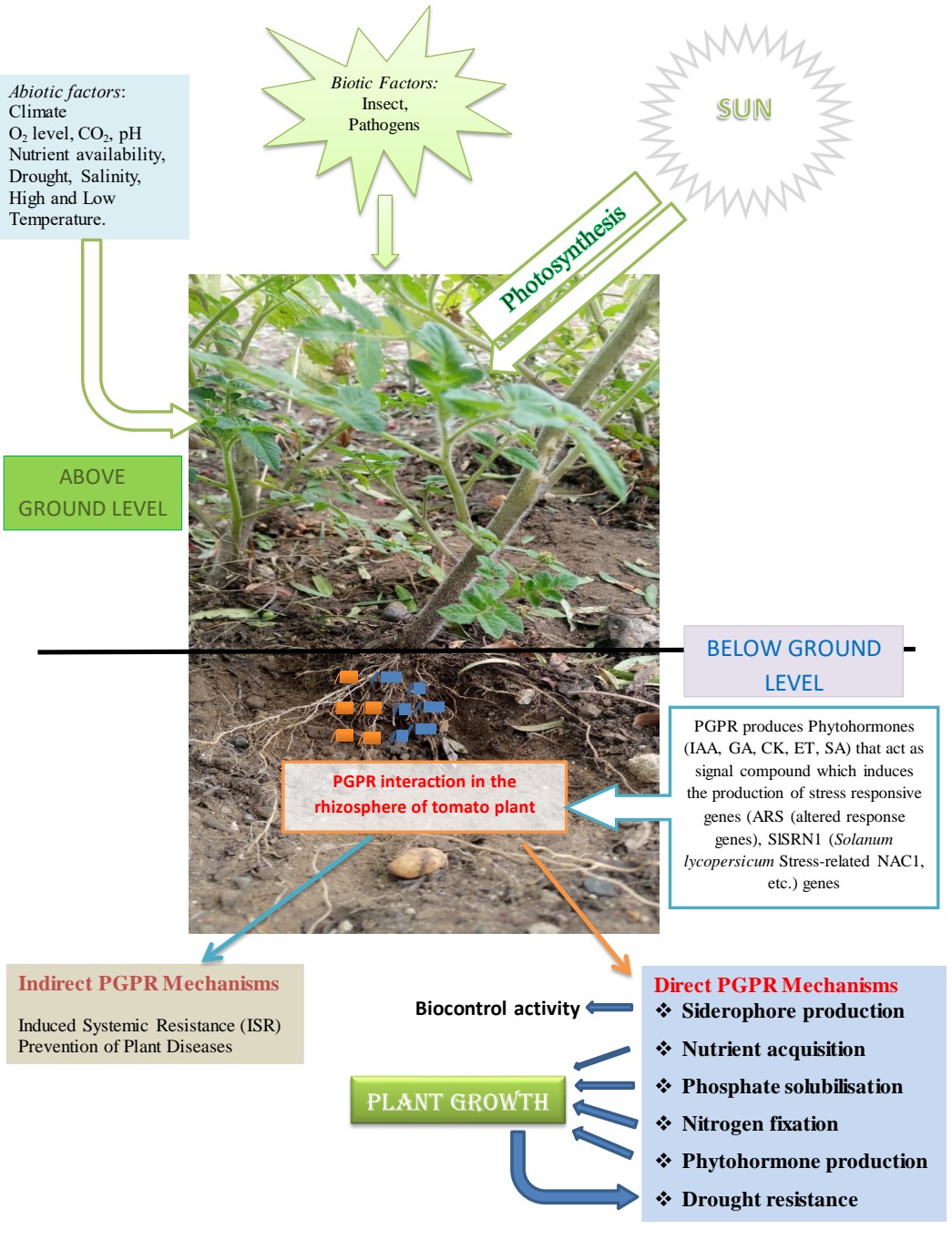

**Figure 1** The potential of PGPR in the rhizosphere of tomato plant.

(*Babalola, Adedayo & Fadiji, 2022*). This rhizospheric soil comprises a high quantitative number of bacteria, unlike those found in bulk soil. The bacteria from the rhizosphere soil mentioned above can yield control and be used for environmentally friendly biotechnological applications (*Santoyo et al., 2021*). The bacteria assist in producing metabolites, which serve as antimicrobials (biological control) against spoilage organisms,

biological remediation, and natural fertilization agents, thus enhancing soil fertility, soil health, crop production, and promoting sustainability of the agricultural environments (*Fasusi, Cruz & Babalola, 2021*).

Tomatoes (*Solanum lycopersicum*) were cultivated globally. It succeeds potatoes that belong to the family of *Solanaceae*. It is widely known to be the second most commercially famous and eatable vegetable fruit (*Włodarczyk, Smolińska & Majak, 2022*). It is utilized by various crop plants to study disease resistance systems' genetics and molecular features. However, the tomato crop is subjected to danger worldwide due to living and non-living factors that result in severe harvest and reduced productivity. It has been reported that tomatoes accommodate up to 200 plant diseases, including fungi, bacteria, nematodes, viruses, and other pervert plants at various phases of development, decreasing the harvest rate and quality of the vegetable (*Shahzad et al., 2021*).

*Ralstonia solanacearum,* a causative agent of bacterial wilt disease, has been the focus of most research on tomato disease (*Xu et al., 2022*). *Zhou et al. (2021)* reported the considerable taxonomic and functional changes between diseased and healthy tomato-associated bacterial populations caused by *R. solanacearum.* The healthy tomato bacterial community interacts more frequently and consistently than the bacterial wilted tomato community, presumably increasing community stability against *R. solanacearum* incursions (*Wei et al., 2019*).

This literature survey evaluates the impact of PGPR in improving the production of tomato fruits through their association with tomato plant roots and the production of various phytohormones. This significantly contributes to plant growth sustainability, activates plant immunity, sustains tolerance of stressors, and aid plant maturation for fruiting. This review examines the potential of PGPR composition to enhance the environmental health status of tomato plants. It also investigates how the rhizosphere soil-inhabiting rhizobacteria vary in agricultural practices and explains how the knowledge obtained will help understand PGPR that promotes the production system.

## Survey methodology

To ensure an inclusive and impartial investigation of literature and to carry out the review's objectives, a comprehensive investigation of published articles on the activity of plant growth-promoting bacteria mechanism of action was employed following the method of *Mayak, Tirosh & Glick (2004)*, *Olanrewaju, Glick & Babalola (2017)*. It is however intended for agriculture, food safety, and sustainability. Search results were gathered and complied employing the online endnote library system. This helped to arranged useful article embedded in the context. However, we appraised the titles, abstracts and the conclusion of the literature to determine the useful ones.

## THE RHIZOSPHERE

This is the soil surrounding the roots of plants and where the interactions of microbial communities and plants take place. In rhizosphere soil, several physical, chemical, and biological features are adhered to modulate planting processes (*Sahu et al., 2019*). It is also the soil habitat of intense interaction between bacteria and plants. Their structure and

functions are influenced by the type and texture of the soil, environmental properties, and plants growing on the soil (*Nwachukwu, Ayangbenro & Babalola, 2021*). The plant root exudates and other rhizodeposits lure good bacteria into the rhizosphere. The plant host brings on the selection pressure due to the advancement of the microbiota present in the rhizosphere, which supports and draws in a circumstantial plant microbiome due to alterations in the spatial relation of the root exudate (*Carrión et al., 2019*).

*Odelade & Babalola (2019)* reported a high total count of bacteria in the rhizosphere soil, unlike the total count of bacteria in bulk (control) soil with no plantation. This is because of high nutrient availability to support the growth of bacteria by root exudates, thereby amounting to a higher microbial population and diversity of the community in the root region that is not the same as those found in the bulk soils. The report was in line with the study of *Kari et al. (2019)*, which revealed the total count of bacteria cells is $10^8 10^{12}$ CFU in 1g of rhizosphere soils. However, the count is more than the bacterial cells in the bulk soil because of exudates produced by the root and rhizodeposition surrounding the roots in the rhizosphere soil. In the soil, bacterial density is in abundance because of the high relative humidity and nutrient availability (*Nwachukwu, Ayangbenro & Babalola, 2021*).

The composition and gathering of the plant-associated microbiota and the physicochemical properties of the soil influence disease outcomes (*Carrión et al., 2019*). However, it is unknown if the microbiota of spoilage organisms of infected tomato plants differs from those of uninfected tomato plants. Furthermore, earlier reports only showed the habitat of bacteria in healthy and diseased tomato crop plants, excluding fungi and other important microbes (*Kwak et al., 2018*; *Wei et al., 2019*). Some fungi, *Trichoderma viride* and *Penicillium chrysogenum* for example have demonstrated the inhibition activity of the spoilage organisms and attribute the arrangement of microbial communities to the plant root (*Omomowo, Adedayo & Omomowo, 2020*). As a result, we need to investigate how bacterial and fungal communities function between healthy and diseased tomatoes.

## Effects of various chemical derivatives on crops

Chemical derivatives have been used for a long time for ameliorating the soil's richness and aid bountiful harvests in agricultural practice. Fertilizers, insecticides, fungicides, and herbicides, among others, have been applied by farmers on farmland but this has raised health concerns for accumulating toxic chemicals in human tissue, animals, and plants (*Ajilogba, Babalola & Ahmad, 2013*). Subsequently, health concerns about accumulating toxic chemicals in human tissue, animals, and plants have arisen. It even pollutes the environment, especially aquatic lives, thereby causing suffocation of water organisms and other health hazards to humanity (*Khalid et al., 2017b*). Chemicals used on farmland entered human circulatory systems by inhalation, oral ingestion, or through the process of diffusion through the skin (*Verla et al., 2019*). Pesticides are mostly known to show long-term persistence in food materials like fruits and vegetables (*Gallo et al., 2020*). In addition, some individuals are hypersensitive or allergic to this chemical already in their system, causing various illnesses like cardiac disease, respiratory disorder, and musculoskeletal weakness (*Khalid et al., 2017*).

Various challenges have been encountered due to the application of synthetic insecticides, herbicides, fungicides, and other chemical derivatives (Lorsban or chlorpyrifos). They have amounted to plant diseases creating microbial resistance genes and are therefore resistant to these chemical derivatives (*Ajilogba & Babalola, 2013*).

Having experienced problems caused by the derivatives mentioned above, the entire globe is trying to produce healthy, ecologically friendly crops without chemicals. An alternative to lessen the application of chemicals in farming systems is to apply microbial inoculants (*Chen et al., 2019*). They are called biofertilizers, biostimulants, or biopesticides and improve the soil's fertility. In addition, they encourage crop growth and prevent spoilage organisms from invading the crop plants. These organisms are primarily in the rhizosphere of healthy growing plants. Their aim is for a natural and ecological equilibrium (*Omomowo, Adedayo & Omomowo, 2020*).

## PGPR effect on tomato productivity and their mode of action

Regarding agricultural practices, PGPR is used for its potential to increase tomato plant development and improve tomato plant protection from various infections and non-living factors like salinity and drought (*Numan et al., 2018*). PGPR is found on the tomato plants' organs such as the root surface attached to or in the soil (and therefore called rhizobacteria), or the endophyte, which is the interior parts of the plant (*Jambon et al., 2018*). They produce innate procedures which promote the nutrient rate of assimilation as a biostimulant and quality of crops (*Emmanuel & Babalola, 2020*). PGPR has been a potential living composition of nano-biological fertilizers that can aid plant growth and development and avoid the development of dependent fungi (*Gouda et al., 2018*).

PGPR assists and encourages the development of the tomato plant through direct or indirect mechanisms (*Berger, Baldermann & Ruppel, 2017*). There are various ways by which PGPR promotes tomato plant growth as observed in Table 1. This procedure can be performed independently or in a group, especially with the rhizobacteria beneficial to the tomato plants. The characters that result in the direct promotion are called the natural mechanism of plant growth. Here beneficial compounds are provided to host plants and produce nutrient assimilation from the rhizosphere. On the other hand, indirect mechanisms are those characters that disallow the operating of one or more spoilage organisms of a tomato plant (*Ajilogba & Babalola, 2013*).

Soils containing microbial communities and huge organic matter require less fertilizer than naturally managed soils (*Mahal et al., 2019*). The huge microbial process is a typical example in soils frequently considered when applying organic nutrient sources. Phytomicrobiome research explains how to show a particular plant-microbe relationship that directly assists plant nutrition (*Vishwakarma et al., 2020*).

According to *Salehi et al. (2019)*, a vegetable crop like tomatoes is a horticultural crop that promotes its consumer's health due to the nature of certain nutrients found in them. Tomatoes contain nutrients and antioxidants, which include oxalic acid and ascorbic acid. These antioxidants in tomatoes are known to neutralise toxic free radicals in the blood circulation, reduce cholesterol levels and prevents high blood pressure (*Mallick, 2021*). When a crop seedling is inoculated with an actinomycete strain of rhizobacterium, the

**Table 1** Rhizobacteria and their various effects on tomato plants.

| Rhizospheric Plant | Rhizobacteria | Effect(s) | Reference |
|---|---|---|---|
| Tomato (*Solanum lycopersicon*) | *Pseudomonas* sp., *Curtobacterium* sp. | They prevent cold stress that inhibits the tomato plantation, development, and productivity of tomatoes, especially by the following organisms: *Pseudomonas, Curtobacterium, Janthinobacterium, Stenotrophomonas, Serratia Brevundimonas, Xanthomonas, Frondihabitans, Arthrobacter, Pseudarthrobacter* | *Vega-Celedón et al. (2021)* |
| | *Burkholderia gladioli* C101 | They produced heat-stable active secondary metabolites that prevent the growth of tomato spoilage organisms *Xanthomonas perforans* | *Shantharaj et al. (2021)* |
| | *Bacillus species* | Isolates TRS-7 and TRS-8 among isolate of *Bacilli* were the best plant growth promoters among the seven isolates, with potential as inoculants to improve the production of tomatoes. | *Kalam, Basu & Podile (2020)* |
| | *Rhizobium sp.* | Rhizobial strains to support and improve the growth of *Solanum lycopersicum* under limited supply of nitrogen | *Zuluaga et al. (2020)* |
| | *Pseudomonas, Bacillus, Azotobacter, Enterobacter, Azospirillum* | These rhizobacteria contribute to the growth of these vegetables like tomatoes, pepper, onion | *Mekonnen & Kibret (2021)* |
| | Actinomycete *Streptomyces sp.* KLBMP5084 | As the biofertilizer these strains can promote the tomato seedlings' growth in salinity stress condition. | *Gong et al. (2020)* |
| | *Proteobacteria, Bacteroide, Actinobacteria* | These are tomato-associated bacterial communities that assist in the production of tomatoes | *Dong et al. (2019)* |

considerable amount of glucose, fructose, nitrate, maleate, zinc, and phosphorus are found embedded in the harvested fruit (*Gouda et al., 2018*).

## PGPR promotes plant growth and resistance

The production of phytohormones by PGPR has important attributes on the growth and health status of the tomato plant (*Vasseur-Coronado et al., 2021*). These hormones are important signaling molecules that control the defense mechanism and growth of tomato plants. The auxin hormone (Indole acetic acid (IAA)) is a spectacular hormone produced in the rhizosphere of a healthy plant (*Poveda et al., 2021*). Phytohormones do exist in synthetic and non-synthetic forms and are sub-divided into five classes based on their sameness and their effect on plants. Some hormones required to regulate the growth of plants are known as synthetic hormones and are classed as chemically, naturally, or organically as produced by PGPR that may be obtained through several processes (*Seenivasagan & Babalola, 2021*). PGPRs produce phytohormones thereby indicating their advantages to the tomato plants and the rhizosphere they inhabit. The basis of phytohormone for the activity of plant growth-promotion of natural biostimulators is ascribed to improving tomato growth (*Kapadia et al., 2021*). Therefore PGPR ameliorates tomato plant growth and as well improves the production of auxins (IAA), gibberellin (GA), and salicylic acid (SA) in plants. Below is concise information of some phytohormones which include IAA,
cytokinin (CK), ethylene (ET), GA, and SA likewise growth regulators like nitric oxide and polyamines produced by tomato and other plants.

### Auxins

PGPR are known to produce auxin which several reports have explained how IAA can be a signaling molecule in IAA-producing and IAA-non-producing microbial species (*Batista et al., 2021*; *Park et al., 2021*; *Uzma, Iqbal & Hasnain, 2022*). These reports express various ideas on the function of IAA in PGPR and their interaction with tomato plants. These phytohormones produced by PGPR affect tomato plants' physiology directly most especially in the root colonization process adopted by PGPR while the association of plant-microbe takes place. IAA acts as an indicating molecule in PGPR, so it influences positive outcomes in the tomato plants, from phytostimulation to immunity of the plant (*Samaras et al., 2021*; *Shahid et al., 2021*). IAA control growth by stimulating cell elongation in stems, carrying out cell division and differentiation, fruit development, formation of roots from cuttings, reduction of lateral branching (apical dominance), and leaf fall (abscission) (*Tan et al., 2021*).

### Gibberellin

GA are phytohormones that have revealed the ability to control growth processes like stem dormancy, elongation, flower development, flowering initiation, germination, fruit development, and leaf and fruit senescence or aging in plants (*Saidi & Hajibarat, 2021*). GA effects on plants brings are somewhat the same as auxins', despite their mode of action are not the same. Dwarfism in plants results as a result of low or no concentration of gibberellin (*Dong et al., 2021*).

### Cytokinins

CKs are other plant hormones that are known to control plant growth and development, like apical development, cell division, root elongation, stomatal behavior, and chloroplast synthesis (*Cavallaro et al., 2022*). They are produced in the root of plants from adenine compounds. They move through the vascular tissue (xylem) to the leaves, fruits, and other parts of the plants where growth and differentiation are required. The introduction of CKs can improve tomato plant and fruits development and as well as the potential to improve pathogenesis in the tomato plant (*Toribio et al., 2021*). CKs also perform a specific role with auxin to reverse senescence in plants by modifying the level of proteins and synthesizing chlorophyll in the leaves to reduce the yellowing of leaves in plants (*Guo et al., 2021*).

### Ethylene

ET is a crucial phytohormone that promotes the ripening and rotting of tomato plant fruits (*Tao et al., 2021*), and the only phytohormone that happens to be a gas. However, it can be produced mostly in every tissue of the plant and can diffuse out of the plant. This procedure induces the stimulation of 1-amino cyclopropane-1-carboxylic acid which is an ethylene precursor and modifies ACC oxidase activity.

### Salicylic acid

SA is also an important phytohormone that belongs to the phenolic group. There are tremendous physiological advantages in plants as a result of their potential to control the growth and development of plants through the following processes; photosynthesis, respiration, transpiration, and the transportation of ions (*Aqeel et al., 2021*). When plants are been exposed to biotic and abiotic stresses, SAs were activated thereby performing various functions like modulation and regulation of numerous responses (*Roeber et al., 2021*). They also can activate and produce various signaling pathways by associating with other phytohormones like ET that perform a significant function in reducing plant stresses.

### Soil microorganisms' effect on tomato plant growth promotion

The narrow soil zone extends on all sides of developing tomato plant roots. It corresponds to a significant area for the activity of microbes in the plant rhizosphere (*De La Fuente Canto et al., 2020*). A large number of taxonomic microbes include prokaryotic organisms (viruses, bacteria, and archaea) and eukaryotic organisms (algae, arthropods, fungi, nematodes, and protozoa) inhabit this soil. Bacteria and fungi contain mostly prevalent units revealing elementary ecological purposes (*Mazière et al., 2021*). PGPR is designated and assists plant development as free-living bacteria inhabiting soil flourish well and competitively inhabit the plant roots (*Basu et al., 2021*).

The diverse bacteria members, an essential part of the microbiota found in the soil, manufacture and liberate various modulatory compounds from the vicinity of the plants' root that assists its growth (*Khoshru et al., 2020*). The contribution of nutrient acquisition enhancement by plants determines plants' health by PGPRs, thereby keeping them safe from phytopathogenic microbes and enhancing resistance to non-living factors (*Backer et al., 2018*). Different genera of PGPR strains possess the potential of biological control activities, promote resistance to foliar spoilage organisms, improve crop yields, enhance nodulation in legumes, and promote the seedlings' occurrence (*Rozier et al., 2017*; *Kalam, Basu & Podile, 2020*). The bacteria are *Acinetobacter, Aeromonas, Agrobacterium, Allorhizobium, Arthrobacter, Azoarcus, Azorhizobium, Azospirillum, Azotobacter, Bacillus, Bradyrhizobium, Burkholderia, Caulobacter, Chromobacterium, Delftia, Enterobacter, Flavobacterium, Frankia, Gluconacetobacter, Klebsiella, Mesorhizobium, Micrococcus, Paenibacillus, Pantoea, Pseudomonas, Rhizobium, Serratia, Streptomyces, Thiobacillus,* and other reported PGPRs (*Ankati & Podie, 2018*; *Kalam, Basu & Podile, 2020*).

Among beneficial microbes in the soil community, bacterial species are the most abundant and helpful. *Saravanan et al. (2020)* gave the following full details of the action of bacteria in the soil: they help stimulate plant growth after the production of the certain phytohormone responsible for the development of plants; they return nutrients to the plants by fixing nitrogen back to the soil; they promote soil structure; they act against spoilage organisms which can destroy the crop plants. Therefore, PGPR is naturally more beneficial to the soil that they colonize.

According to *Basu et al. (2021)*, a putative PGPR strain possesses plant promoting-growth characteristics and promotes growth once it has been inoculated into a plant. Below are the features of PGPRs in the rhizosphere soil of a tomato as stated by

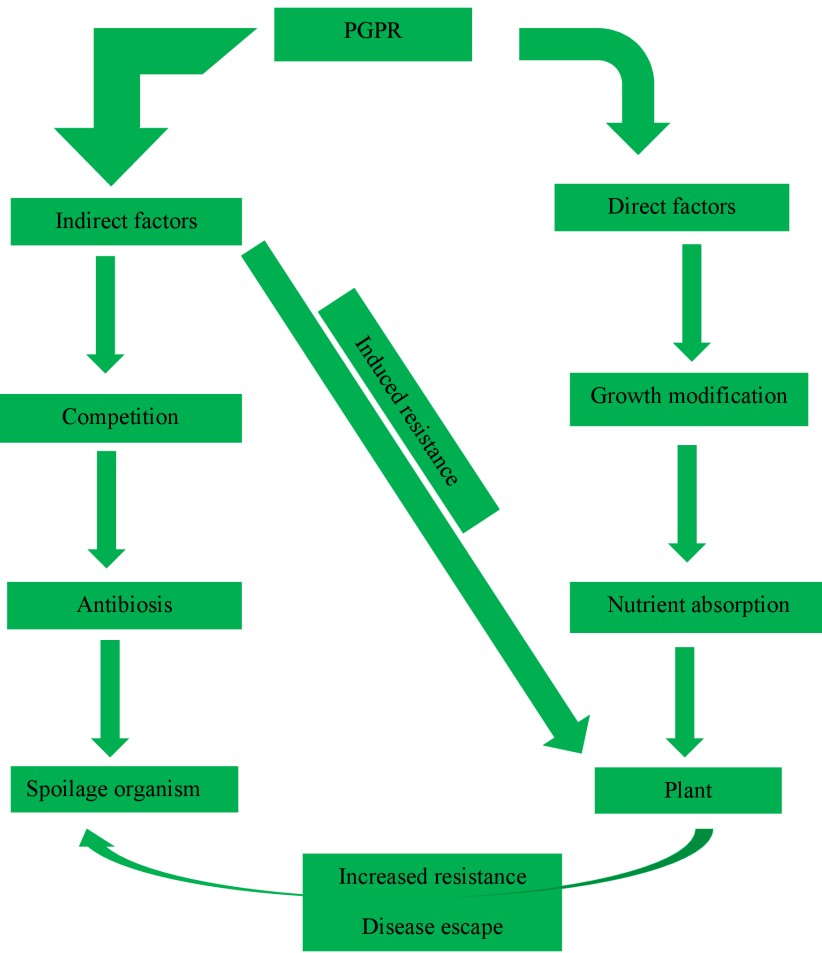

**Figure 2** **Various roles and functions carried out by PGPR in the rhizosphere.**

*Guerrieri et al. (2020)*: they are eco-friendly and rhizosphere-competent; they promote plant growth and development; they exhibit a broad spectrum of actions; physical fractures like high temperature, oxidants, radiations, desiccation, should be tolerated by the PGPRs; there is positive interaction between them and other bacteria in the soil; they should be capable of adhering to the plant root after being inoculated in the rhizosphere, and they should be able to demonstrate better competitive skills over rhizobacterial communities already existing in the rhizosphere as shown in Fig. 2.

## MACROELEMENT SOLUBILIZATION IN THE RHIZOSPHERE

Macroelements, also known as macronutrients, are elements or nutrients required by plants in large quantities. The rhizosphere contains macroelements like nitrogen, calcium, magnesium, potassium, phosphorus, and sulfur (*Kadyampakeni & Chinyukwi, 2021*). PGPR assists in the solubilization of various elements required by the plant (*Wenzel, 2009*). More of these organisms and their fixing of these significant elements in the rhizosphere are explained in Table 2.

**Table 2** Macroelement solubilization and activity of rhizobacteria in tomato and other crop plant rhizosphere.

| Macroelements | Tomato /Crop plants | Bacteria | Effect | Reference |
|---|---|---|---|---|
| Nitrogen | Beans (*Phaseolus vulgaris*) and okra (*Abelmoschus esculentus*) | *Pseudomonas alcaliphila, Pseudomonas hunanensis, Streptomyces laurentii, Sinorhizobium sp.,* and *Bacillus safensis* | The bacteria significantly promoted the growth of the root of beans (*Phaseolus vulgaris L*) and can be used to manufacture biofertilizer | *AlAli, Khalifa & Al-Malki (2021)* |
| Potassium | Tomato (*Solanum lycopersicum*) | *Enterobacter hormaechei* (MF957335) | These bacteria are potassium solubilizing bacteria. They are great use for plant growth under saline condition, thereby contributing to the growth of the tomato plant and root elongation as a result of potassium fixation. | *Ranawat, Mishra & Singh (2021)* |
| Magnesium | Rice (*Oryzae sativa*) | *Alcaligenes* species | This bacteria displayed attributes at a different level of magnesium salt concentration which favors rice growth. | *Fatima et al. (2020)* |
| Iron | Chickpea (*Cicer arietinum*) | *Azotobacter chroococcum* (AU-1), *Bacillus subtilis* (AU-2), *Pseudomonas aeruginosa* (AU-3), and *Bacillus pumilis* (AU-4) | These rhizobacteria showed plant growth-promoting characters and iron chelating siderophores, allowing promotion of the development and production of chickpea plants under normal conditions. | *Pandey, Gupta & Ramawat (2019)* |
| Sodium | Barley (*Hordeum vulgare L.*) | *Bacillus mojavensis* S1, *B. pumilus* S2, and *Pseudomonas fluorescens* S3 | Sodium concentrations promoted leaf water ability, and the strain S1 kept it in line to attribute ideas | *Mahmoud et al. (2020)* |
| Phosphorus | Tomato (*Solanum lycopersicum*) | *Bacillus safensis* B23, *Bacillus aryabhattai* B29, *Bacillus subtilis* B18, *Bacillus subtilis* B25, *Pseudomonas moraviensis* B6 and *Bacillus simplex* B19 | The rhizobacteria solubilized phosphate and further improve tomato plant growth | *Cochard et al. (2022)* |
| Sulfur | Tomato (*Solanum lycopersicum*), Orange (*Citrus sinensis*) | *Bacillus, Klebsiella, Pseudomonas, Azobacter, Enterobacter, Serratia, Variovorax, and Azospirillum* | The rhizobacteria play an essential duty in sulfur cycling, thereby increasing the production of tomatoes. | *Rai et al. (2020)* |

Biofertilizers are substances that contain living microorganisms that, when applied to plant surfaces, or soil, populate the rhizosphere or the interior of the plant thereby promoting growth by increasing the supply or availability of primary nutrients to the host plant. They are also regarded as substances that contain microbes that help in the plant nutrient acquisition process through increasing surface areas like plant roots, hydrogen cyanide production, siderophore production, nitrogen fixation, and P-solubilization (*Singh et al., 2019b*). Therefore, improving soil microbial activity can make tomato crops available with their nutritional values (*Ye et al., 2020*). Although through this procedure, PGPR has many advantages for plants and helps in the accumulation of both minor elements (Zn, Co, Mn, *etc.*) and significant elements (Na, K, Mg, N, *etc.*) (*Ramakrishna, Yadav & Li, 2019*). Significant elements like Mg and K are the most important elements that increase the standard of plants (*Ceccanti et al., 2021*).

Potassium-solubilizing bacteria (KSB) are bacteria that can increase nutrient availability in the soil by producing K from non-soluble materials (*Numan et al., 2018*). Bacteria in organic forms release the chemical compounds tartaric acids, citric acid, gluconic acid, succinic acid, oxalic acid, malic acid, and 2-ketogluconic acid. They can dissolve rocky K ions or chelate compounds, releasing K into the surrounding soil for the plants to assimilate (*Mason-D'croz et al., 2019*). Inoculating tomato crops with KSB, improves the presence of K in the tomato plant's rhizosphere, thereby producing an abundant crop harvest (*Raji & Thangavelu, 2021*).

One of the natural processes required for plant growth includes phosphate solubilization. Secondary metabolites like alkaloids, terpenes, phenolics, lipids, and saponins present in phosphate solubilizing bacteria (PSB), assist in promoting the taste and encouraging health characteristics of food crops (*Ramakrishna, Yadav & Li, 2019*).

## Solubilisation of phosphorus

For a healthy plant to grow and develop, phosphorus is one of the main elements required. The element is readily available and found in the soil. Since this element is present primarily in an insoluble form, rhizobacteria in the soil help solubilize the phosphorus, making it usable by plants by accumulation and transformation of phosphate to plant roots. The following symptoms are present in phosphorus-deficient plants: purple coloration of the underside of the older leaves due to accumulation of anthocyanin pigment (*Pongrac et al., 2020*).

The tomato plant absorbs phosphate quickly due to the high absorption surface area gradient of the plant's root. Rhizobacteria are known to solubilize insoluble phosphate, which is why they are culturable on growth media in the laboratory by showing the area of phosphate solubilization (*Santoro et al., 2021*). These media contain various constituents like aluminum, iron, tricalcium phosphate, rock phosphate, and hydroxylapatite. The following bacteria can solubilize phosphorus in the soil: *Burkholderia, Azobacter, Pseudomonas, Bacilli, Enterobacter, Citrobacter, Pantoea,* among others (*Kaur et al., 2017*). During the solubilization of insoluble organic phosphorus, two enzymes are involved in the process: phytase and phosphatase. Bacteria produce organic compounds like gluconate,

citrate, ketogluconate, tartrate, lactate, oxalate, which help solubilize inorganic phosphate (*Babalola et al., 2021*).

### Nitrogen fixation

Nitrogen fixation can be defined as the process by which nitrogen present in the atmosphere is converted to ammonia by nitrogen-fixing bacteria for plants to utilize (*Yilihamu et al., 2020*). Nitrogen is among the essential ingredients needed by the tomato plant. It increases leaves' size and quality. Its deficiency in plants results in limited growth of plants and yellowing of plant leaves (chlorosis) (*Caradonia et al., 2019*). Certain microbes add nitrogen to biofertilizers and have become a significant concern for researchers due to their environmentally friendly nature. Certain bacterial strains help fix atmospheric nitrogen and ensure its availability for tomato plant utilization (*Masood, Zhao & Shen, 2020*). Some examples are *Enterobacter, Bacillus, Azobacter, Klebsiella, Serratia, Azospirillum, Arthrobacter, Gluconacetobacter,* and *Pseudomonas.* This microorganism forms a symbiotic relationship with plants by adding the atmospheric nitrogen, and the plant, in return, houses them in their rhizospheric soil (*Rozier et al., 2017*). *Cyanobacteria* and *Azolla* can also implement required nitrogen by plants, as reported by (*Akhtar et al., 2021*).

### Potassium solubilization

Potassium is one of the macro components needed by plants. Chemically, it can be used to produce NPK fertilizer. However, tomato plants absorb potassium as an ion that can readily be leached and lost through soil runoff (*Sardans & Peñuelas, 2021*). This element is required in plants to promote the formation of sugar for protein synthesis, root growth, and cell division in plants. A significant deficiency experienced on plants lacking potassium is leaf edge chlorosis. The economic importance of the defect is that the chlorosis is irreversible even if one adds the potassium later.

### Magnesium solubilisation

Magnesium is the primary element required for the structural component of chlorophyll. A tomato plant needs it to promote the function of plant enzymes to produce carbohydrates, fat, and nutrient absorption regulation (*Kwon et al., 2019*). Magnesium deficiency in plants leads to chlorosis in tomato leaves, and severe cases result in stunted growth (*Bang et al., 2021*). The PGPR was reported to produce several metabolites including siderophores, organic acids, and growth hormones, which promote solubilization of iron and magnesium to the plant (*Asad et al., 2019*).

## VARIOUS ASSOCIATIONS OCCURRING BETWEEN MICROORGANISMS AND TOMATO PLANTS

Microbes form an interrelationship with tomato plants in their habitat, and such relationships are very promising. These interrelationships create the protection of the tomato plants against spoilage organisms and manufacture materials needed for the plants while others are harmful to the plant (*Bordewijk & Schifferstein, 2020*). Microorganisms' value to the soil through interactions in the rhizosphere are materialistic in promoting and increasing tomato production in the agricultural sector (*Verma et al., 2018*).

*Igiehon & Babalola (2017)* explained more about Soybean (*Glycine max*) plantation. It is a leguminous plant which is member of the order *Fabales* and family *Leguminosae*. It can produce a mutualistic relationship with rhizospheric microorganisms. This literature has proclaimed that these mutualistic microorganisms support their hosts in the habitat they colonized.

Knowing more about relationships among microorganisms in the rhizosphere soil is a quick procedure for farming processes that do not use chemical fertilisers that are detrimental to plants and animals consuming them. Below are some of the associations that take place among organisms:

## Parasitic association

Parasites live and feed on other organisms called the host, and this host suffers due to the organism feeding on it. Fungal parasites explain the parasitic relationship between *Stachybotrys elegans* and *R. solani,* which shows the concentrations of different secondary metabolites (*Latz et al., 2018*; *Carroll et al., 2021*). *Meloidogyne spp.* is a parasitic nematode of tomato plants, most economically and globally significant. It is challenging to eradicate and control the parasites *Meloidogyne spp.,* because of the parasite infection on the tomato plant. Chemical nematicides contribute to the high toxicity of the plant. Rhizobacteria have been a prominent alternative to control these parasites without negative impact on the tomato plant, animals, and other organisms feeding on it. *Bacillus spp.* was effective and acted as biocontrol agents for plant pests and diseases (*Chen et al., 2020*) because it has many functions, including fixing phosphate, increasing the plant's growth, and much more (*Franco-Sierra et al., 2020*). According to *Habazar et al. (2021)*, *Bacillus spp.* is a rhizobacterium employed to control *Meloidogyne spp.*, to improve the growth and cultivation of tomatoes. Bacterial species *Pasteuria penetrans* also act on nematodes to decrease root-knots growth through a parasitic interrelation. This bacterium multiplies in infected nematodes, killing them or causing infertility among those that survive the action. Once giant spores produced by bacteria are attached to the growing nematodes present in the rhizosphere, the movement and penetration of these nematodes are reduced (*Heinrichs & Muniappan, 2018*). In particular, plants have promoted their defense mechanism to fight invading spoilage organisms (*Köhl, Kolnaar & Ravensberg, 2019*). Some plant genes are RNA-seq responsible for defense against a plant root spoilage organism, *Verticillium dahlia* (*Berne et al., 2020*). They produced a plant-based signal transduction pathway web, which was initiated to acknowledge elicitors with safety indication materials and Pathogen-Associated Molecular Patterns that observe microbes like those that essentially relate to the roots of the plant in their soil habitat. The association of microorganisms obtained in the greenhouse field can reveal their potential thereby encouraging the isolation of beneficial soil microbes that will possess parasitic and biological control characteristics on plant spoilage organisms (*Mills, Ross & Hill, 2017*).

## Symbiotic association

*Igiehon & Babalola (2018)* reported that symbiosis is a mutual relationship or association that involves two or more organisms to benefit both. A typical example of this association

is the nitrogen-fixing bacteria (rhizobacteria) and roots of tomato plants (*Masood, Zhao & Shen, 2020*), as observed in Fig. 2. Rhizobacteria manufactured certain materials to increase the development of their host plants. They also assist in fixing nitrogen to plants and obtain a mutualistic relationship while also yielding to living and nonliving factors (*Devi et al., 2020*). Compounds like 2, 3-butanediol, among others (volatile organic chemical), are released by some rhizobacteria that elicit induced systemic resistance (ISR) in the plant. These compounds diffuse diketopiperazines rhizosphere interrelated bacilli, producing lipopeptides, polyketides, biosurfactants, and siderophores with prominent signal factors that are involved in molecular cross-talks between members of plant microbiota (*Andrić et al., 2021*).

The compounds like butanediol are released by symbiont *B. subtilis,* which is known to inspire induced systemic resistance by tempering the transcription of Na $^+$conveyer in plants (*Oleńska et al., 2020*). Likewise, rhizobacteria are known to produce some materials that promote tomato root penetration during growth by reducing the penetration of the primary root and promoting the formation of the distal root (*Khanna et al., 2019*). Some bacteria, together with fungi, liberate auxin hormone that comes in contact with the signs of this material in the root region (*Singh et al., 2019a*). Yet, in the root endophyte, auxin derivatives released by *Piriformo sporaindica* do not showcase action in the root development of a barley plant but are primarily available for parasitic infection occurring at the root of the plant. These bacteria and fungi found at the root also initiate compounds (dimethyl disulfide and pyocyanin) that control the growth of plant roots by creating the procedure of signalling auxin (*Khatoon et al., 2020*).

The mutualistic relationship often occurs in the rhizosphere between particular plants and microbes. These mutual relationships also occur between the microorganisms inhabiting the soil environment. *Igiehon & Babalola (2017)* reported the association of beneficial bacteria and arbuscular mycorrhizal (AMF) as a mutual relationship because the bacteria assist the fungi in a profound reciprocal relationship. At the same time, AMF improves bacterial intrusion ability and differences, though other advantages can be achieved in this relationship. Interactions do occur between fungi (*Rhizopus*) and bacteria (*Burkholderia*) which was regarded as a symbiotic association in the rhizosphere of tomatoes (*Zhang et al., 2021*). In the absence of the bacteria, the fungi will not produce spores, which reveals that both organisms depend on each other for reproduction and survival, *i.e.*, the fungi live on the compound produced by the bacteria (*Del Barrio-Duque et al., 2020*).

## Antagonistic association

This association is another form of interrelation between two or more organisms, either identical or different species. One organism dominates the other and prevents it from carrying out characteristics of life, including growth and feeding. In the tomato rhizosphere, antagonists produced specific chemicals which harmed other organisms. They produced enzymes such as lipases, protease, cellulases, and chitinases. These enzymes are the organic catalyst that can destroy or break down the cell walls of fungal spoilage organisms (*Karthika, Varghese & Jisha, 2020*). The rhizobacteria actions biologically controlled the

spoilage organisms, which produced these materials to inhibit their features in the tomato rhizosphere soil (*Guerrieri et al., 2020*). Some microbes show important features inhibiting soft rot infection caused by fungi *Rhizoctonia solani* as observed in Table 3. These microbes include *Proteobacteria, Firmicutes,* and *Actinobacteria* (*Piechulla, Lemfack & Kai, 2017*). The fact obtained later explained how particular interrelations are essential to disease suppression. Inhibitory activity of bacteria to destroy *R. solani* on lettuce showed that bacteria possessed limited effect on rhizobacteria and endophytic fungi (*Glick, 2020a*). *Zheng et al. (2018)* gave another instance on the use of biological control bacteria on lettuce that inhibits the action of *R. solani* by fungal and bacterial species found on lettuce. This biocontrol mechanism of action explained the unexpected outcomes which were being explained due to the unfavorable effect on AMF. However, beneficial bacteria like some fluorescent *Pseudomonas* produce 2,4-diacetylphloroglucinol, an antifungal compound that was not injurious to AMF *Glomus mosseae* instead easily promoted inhabitation of root by mutualistic fungus species (*Kabdwal et al., 2019*). The fungal species may support the production of mycorrhizal apart from bacteria when they live on other fungal species. This shows that some fungi live on other species of fungi. Therefore, it is necessary to produce abstraction of fungus (mycorrhizal) for helping bacteria. This was because some species of fungi aider act against fungi that are not helpful and was noticed to promote mycorrhizal production (*Giovannini et al., 2020*).

Therefore, the quality of antifungals can be a reason for an antagonistic selection as biological control agents in the future. *Rhizobium* and *Bacillus*, among other rhizobacteria, are known to liberate siderophores which inhibit spoilage organisms from acquiring iron from the neighboring surroundings and thereby affecting spoilage organisms' existence (*Lurthy et al., 2020*). This culminates in promoting plant growth and productivity.

*Waghunde et al. (2021)* reported that *B. amyloliquefaciens, M. oleovorans, A. xylanus,* and *S. inulinus* have shown high growth inhibition against fungi pathogens. *Tiwari et al. (2021)* explained how *Bacillus subtilis* was used to prevent the development of *Aspergillus flavus* and the poisonous aflatoxin the fungi produced on the farmland and while in storage. According to *Alori & Babalola (2018a)*, various microorganisms, including *Pseudomonads, Mitsuaria* sp, and *Rhizobia,* produces biological control mechanism against spoilage organisms, with the latter suppressing *Pythium* disease, the former inhibiting *Fusarium* wilt and the mid-on bacteria reducing leaf spot of disease plant and more shown in Table 3.

## BACTERIA AND FUNGI AS BIOLOGICAL CONTROL AGENTS

Biopesticides were defined as a group of microbes that show antibacterial and antifungal procedures (*Rani et al., 2017*). In biological control technology used in the agricultural process, microorganic bioinoculants play special features. The potential action of biocontrol agents worked out by most bioinoculants could be characterized by the production of extracellular hydrolytic enzymes and secondary metabolites that can eradicate tomato plant pathogens at a minimum inhibitory concentration, and competition for nutrients (*Lau et al., 2020*), while others influence defense approach like systemic acquired resistance in the

**Table 3  Rhizobacteria as biocontrol agents in tomato and other crop plant rhizosphere.**

| Biocontrol agents | Rhizosphere organisms | Effect on plants | Reference |
|---|---|---|---|
| Rhizoacteria | *Proteobacteria, Actinobacteria, Firmicutes, Acidobacteria, Gemmatimonadetes,* | Biological control of tobacco bacterial wilt (*Ralstonia solanacearum*) | *Hu et al. (2021)* |
| | *Pseudomonas species* | Biological control of *Fusarium oxysporum f. sp. cepae* (FOC) obtained from onion (*Allium cepa*) rhizosphere | *Bektas & Kusek (2021)* |
| | *Bacillus subtilis* K4-4 and GH3-8, *Stenotrophomonas, Sphingobacterium* genus | *Neocosmospora (Fusarium) solani* is a fungi disease affecting orange (*Citrus sinensis*) biologically controlled fruit controlled by the cited bacteria | *Ezrari et al. (2021)* |
| | *Pseudomonas* species and *Serratia* species | The organism reduced egg hatching and promoted mortality rate *in vitro* | *Abd El-Aal et al. (2021)* |
| | *Streptomyces werraensis* F3 | The PGPR was isolated from the rhizosphere of ginseng and was analyzed for antifungal properties against ginseng root rot | *Qi et al. (2021)* |
| | *Alcaligenes faecalis* and *Acinetobacter sp.* | *Clavibacter michiganensis* caused tomato bacteria canker to be controlled biologically by the rhizosphere organisms | *Oloyede et al. (2021)* |
| | *Streptomyces* species | PGPR can biocontrol pathogenic fungi, *Fusarium oxysporum* causing root rot | *Sari, Nawangsih & Wahyudi (2021)* |
| | *Bacillus safensis* RF69, *Bacillus species* RP103 and *Bacillus* species RP242 | Biological control activities have proven to be effective in controlling maize plant spoilage organisms (*Fusarium verticillioides*) | *Einloft et al. (2021)* |

host plants species. The damage from the spoilage organisms to the plants can be reduced by organisms that can arrange the steps of hormones in plant-like gibberellin, cytokine, ethylene, and auxin (*Alori & Babalola, 2018b*). Bioinoculants shown in Fig. 3 produce beneficial effects on plant crop to control bacterial and fungal diseases. The biological control potential of some bioinoculants can be fully explained in herbicidal activity, which is primarily found in mycoherbicide of velvetleaf (*Colletotrichum coccodes*), Striga, and biofungicide of *Fusarium* spp. *Trichoderma harzianum* is a fungus producing volatile antibiotics that suppress spoilage on orange fruits (*Omomowo, Adedayo & Omomowo, 2020*).

According to *Omomowo, Adedayo & Omomowo (2020)*, *Trichoderma viride* and *Penicillium chrysogenum* are biocontrol agents that inhibited the pathogens of orange fruits *Aspergillus niger, Aspergillus fumigatus, Fusarium oxysporum, Penicillium digitatum* and *A. wentii* in dual culture analysis. *Waghunde et al. (2021)* reported that *B. amyloliquefaciens, M. oleovorans, A. xylanus,* and *S. inulinus* have shown high growth inhibition against fungi pathogens.

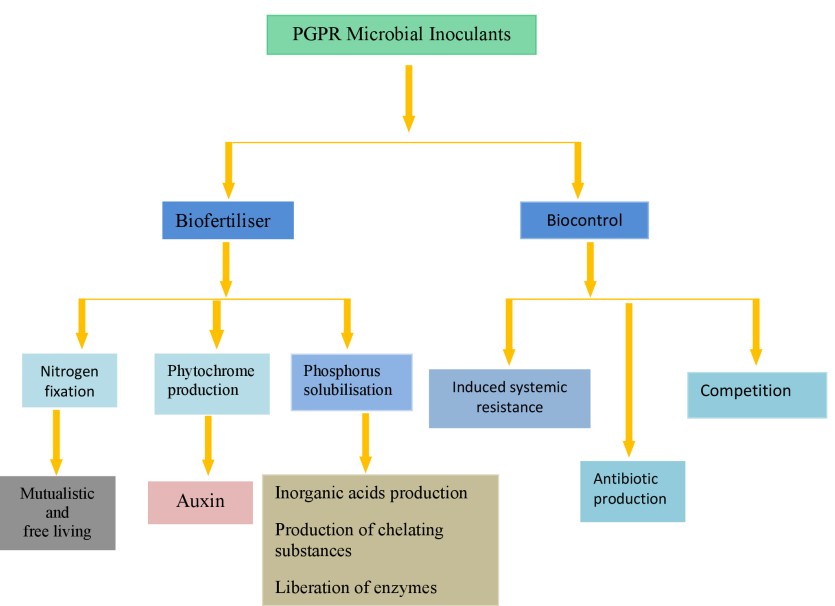

**Figure 3** Diagrammatic summary of some microbial inoculants in agriculture.

# CONCLUSION AND PROSPECT

Nutrient uptake occurs through the particular organ of the plant called the root, which is substantial in the plant's interrelation with PGPR. This review showed the action of PGPR and their relevant functions in the tomato plant. They are known for improving plant growth, fixing nitrate, phosphate, and other essential elements in the soil, bearing against an invasion of spoilage organisms and producing compounds like phytohormones, ammonia and other compounds to induce resistance against pathogens, promote growth and improve health status of tomato plant. They improve crop yield as a result of their biological control activity, thereby reducing the application chemical pesticide that is harmful to human and animal health, and polluting the environment. Rhizobacteria promote the tomato plant's health by improving plant growth resulting in a bountiful harvest and crop quality. However, it is recommended that more research should be done on rhizobacteria and their interrelation with other crop plants and microbiota members to see how they can improve those crop productions. More research should also be carried out on microbes present in the rhizosphere, their relationship with one another, and the plants at their root region to find out how to modify them to be more effective in tomato production.

**Abbreviations/Search words**

| | |
|---|---|
| **AMF** | Arbuscular mycorrhizal |
| **Biocontrol** | Biological control |
| **Co** | Cobalt |
| **ISR** | Induced systemic resistance |
| **K** | Potassium |

| KSB | Potassium-solubilizing bacteria |
| Mn | Manganese |
| Mg | Magnesium |
| N | Nitrogen |
| P | Phosphorus |
| PSB | Phosphate solubilizing bacteria |
| PGPR | Plant growth-promoting bacteria |
| Zn | Zinc |

### Funding

The National Research Foundation of South Africa grants (UID123634 and UID132595). The funders had no role in study design, data collection and analysis, decision to publish, or preparation of the manuscript.

### Grant Disclosures

The following grant information was disclosed by the authors:
National Research Foundation of South Africa: UID123634, UID132595.

### Competing Interests

The authors declare there are no competing interests.

### Author Contributions

- Afeez Adesina Adedayo conceived and designed the experiments, performed the experiments, analyzed the data, prepared figures and/or tables, authored or reviewed drafts of the article, and approved the final draft.
- Olubukola Oluranti Babalola conceived and designed the experiments, analyzed the data, authored or reviewed drafts of the article, and approved the final draft.
- Claire Prigent-Combaret analyzed the data, authored or reviewed drafts of the article, and approved the final draft.
- Cristina Cruz analyzed the data, authored or reviewed drafts of the article, and approved the final draft.
- Marius Stefan analyzed the data, authored or reviewed drafts of the article, and approved the final draft.
- Funso Kutu analyzed the data, authored or reviewed drafts of the article, and approved the final draft.
- Bernard R. Glick analyzed the data, authored or reviewed drafts of the article, and approved the final draft.

### Data Availability

The raw data is available in the tables.

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
