# Peer review of "The application of plant growth-promoting rhizobacteria in Solanum lycopersicum production in the agricultural system: a review"

_PeerJ, doi:10.7717/peerj.13405_

## Round 0.1 · original submission · Major Revisions

Dear Authors,

The review process is now completed.
All reports have advised that your manuscript, even presents intersting results, needs deep modifications that should be addressed accurately.
The manuscript lacks recent refrences and the results and discussion need to be modified and structurated differently.

Reviewer 3 has suggested that you cite specific references. You are welcome to add it/them if you believe they are relevant. However, you are not required to include these citations, and if you do not include them, this will not influence my decision.

Reviewer 1 ·

Basic reporting

The title of the reviewed article suggests that it should be focused on Solanum lycopersicum production, problems, and the role of PGPR in addressing the issue and improving fruit yield. The majority of the study is devoted to generic rhizosphere and rhizobacteria information. All sections describe what PGPR can do, as well as their essential features and functions, however there is no correlation between tomato growth and mechanism of action.

The introduction is irrelevant, sentences are abrupt lack of consistency and not encouraging for the reader

The provided information is too general and basic , the author used obvious facts of filed as information e.g line 50-69, 149-158, 180-184, 239-242 and many others, Similarly, the subsections lack the focus of the main review aim, as it is stated that phytohomrnones play a function, but that it requires a certain mode of action and hormone contribution. Similarly, there is no information on how PGPR promotes plant growth and resistance through altering plant hormones, plant stimulants, or nutrient availability.

The ambiguous use of English language through out the document, sentences are poor in grammar and connection. The inappropriate use of words and terms in the manuscript as, 290“implement nutrient availability”, 294: plant to ingest nutrient, 311: tomato absorb phosphate quickly due to because of the high absorption surface area gradient, Line 282, “Biofertilizers are simply microbes that help the plant nutrient acquisition process through increasing surface areas like plant roots” not so correct,,, microbes are not synonym of biofertilizer these are min component of bio fertilizer and microbes do not increase surface area of plant root

It lacks the previous findings related to tomato plant and growth most of the cited literature is only about pgpr

The presented figures are too generalized to be presented in updated scientific review.

Experimental design

Article fails to meet the prospect aim of review.
The method opted for review selection should target to tomato plant association to PGPR or vice versa to improve plant growth resistance fruit yield and viablity rather than just PGPR qualities that are well-known presently.

Validity of the findings

As the review is too generalised lack the novelty, it brings no additional valid and comprehensive information to the reader.

Review must have latest information about specified topic, and have collection of valid findings in focused issue

Additional comments

The manuscript requires careful consideration in terms of English proficiency, connection, consistency and focus in content. It requires to remove so many redundant details and add precise and rational work from the field

Reviewer 2 ·

Basic reporting

Is the review of broad and cross-disciplinary interest and within the scope of the journal?
Yes
Has the field been reviewed recently? If so, is there a good reason for this review (different point of view, accessible to a different audience, etc.)?
No comments
Does the Introduction adequately introduce the subject and make it clear who the audience is/what the motivation is?
no comments

Experimental design

Is the Survey Methodology consistent with a comprehensive, unbiased coverage of the subject? If not, what is missing?
Lack of novelty and presentation

Are sources adequately cited? Quoted or paraphrased as appropriate?
Almost yes

Is the review organized logically into coherent paragraphs/subsections?
Yes

Validity of the findings

Impact and novelty not assessed. Meaningful replication encouraged where rationale & benefit to literature is clearly stated.
No comments
Is there a well developed and supported argument that meets the goals set out in the Introduction?
no

Additional comments

Comments to Authors:
Manuscript ID: #70929
The manuscript titled “The application of plant growth-promoting rhizobacteria in Solanum lycopersicum production in the agricultural system: A review” submitted by Adedayo et al. summarizes the role PGPR application in tomato plant growth and development. They started with the introduction about the importance of soil health and quality on plant growth and development. Further they talked about pathogens such as Ralstonia solanacearum that limits the tomato yield. In next section they discuss the properties of rhizosphere. Here they talked about chemical derivatives and its negative impact on health followed by the mode of action of PGPR on tomato production. After that, availability and role of macroelements found in rhizosphere like nitrogen, calcium, magnesium, potassium, phosphorus, and sulphur has been explained. In the end they discussed about biological control technology to control bacterial and fungal infection on tomato plants.
Overall, the concept of the manuscript is quite interesting but it is poorly presented. Also, it lacks novelty and there are several flaws in the manuscript that need to be resolved.

 In abstract section: lines 35-36; rhizobacteria that promote the growth and development of tomato plants are referred to as plant growth-promoting bacteria (PGPR). This sentence may confuse to the readers. Please remove tomato from this sentence.
 Lines 42-43: What about other essential nutrients?
 Please check and re-write the sentence of lines 43, 59-61, 73, 92-93, 120-121, 230.
 Lines 59-61: In addition, the following soil factors affect plant growth: water, phytopathogens and parasites, weed seed pools, water, nutrients and dissolved oxygen concentration (Patil & Fauquet 2021). Sentence is not clear please re-write it.
 Line 60: Repetition of “water”. Please remove it.
 Line 69: Figure 1 was cited in introduction section but its legend was presented in another section. Authors should move figure to introduction section after the paragraph where it was first cited.
 Figure 1 is very general representation. There is nothing new for the reader. Please provide the mechanism of action of plant growth promotion in tomato. Authors should show the production of hormones, nutrients by PGPR and their utilization by plants in schematic way. You may also present the name of stress responsive genes in tomato which are induced by PGPR. In this figure 1, abiotic factor: climate, sun, o2 level etc have been mentioned. There are other important abiotic factors such drought, salinity, high and low temperature etc that should be mentioned. Overall, figure should be improved. Also, authors need to properly elaborate Figure 1 in the legend section.
 In the abbreviation section, the authors need to capitalize the first letter and delete the other information. Lines 135-136 are not abbreviations. Also be consistent with the capitalization of the abbreviations used.
 Line 174: Authors wrote “some fungi, for example” but did not mention the name of fungi. What are those fungi?
 Line 187: Please replace “diffusing” with diffusion.
 Table and figures in the manuscript should be presented where authors cited them. Table 2 was cited in line 281 but presented after line 302. Same happened with table 3. Please check and fix it.
 The English language should be improved so that audience could understand the message. Please check and improve lines 104, 122-123, 149, 152-154, 164, 176-177, 192-195, 260, 264-265, 276-277, 284-285, 298, 307, 322-323,390, 425, 465-466, 473, 477,501-502.
 Line 306-308: Since this element is present primarily in insoluble and soluble organic forms, rhizobacteria in the soil help solubilize the phosphorus, making it usable by plants. If soluble is already present then what is the need of rhizobacteria? It would be great if author could explain how rhizobacteria help to solubilize the insoluble phosphate?
 Texts written in lines 462-466 is repeated in lines 489-493. Please check and correct accordingly.
 Authors need to double check all the references, for example, italicized the species (L-532, 577, 597, 732,) and remove extra brackets (L-550).
 In reference cites in lines 635-636 need to mention the location of the publisher.
 In table 2, somewhere authors have used bacterial name whereas somewhere they used only strain names. Please write bacterial names which their strain names properly.

Overall, the manuscript is poorly written and weak presentation. It may be consider after minor revision if authors would be able to justify the comments successfully and improved it.

Reviewer 3 ·

Basic reporting

The current review falls within the biological and important area of research now a days

Experimental design

Literature survey is ok

Validity of the findings

The findings from this review will be beneficial to the readers

Additional comments

The current study entitled “The application of plant growth-promoting rhizobacteria in Solanum lycopersicum production in the agricultural system: A review” is good. For a better understanding in-depth, it is a need for time to work on this topic. Furthermore, the achievement of potential benefits by using current technology is also dependent on the extensive research work for more exploration. Although the literature is well organized, yet I suggest modification due to the following deficiencies.
Major Concerns
• Systematic abstract is missing. Introduce the need for study in 1-2 lines.
• Please give a clear-cut point problem source as a problem statement that is tackled in the current study.
• Give logical reason for the selection of current strategy.
• Quantitative data is also important to support your conclusion. Would you please provide some quantitative data in terms of percentage significant increase or decrease in the abstract?
• Please provide a conclusive conclusion with is withdrawn through research in a single line.
• Please conclude with a statement that shows a knowledge gap covered, potential beneficiaries and specific recommendations as well.
• Give future prospective in a single line.
• As per standard suggestions, please avoid using title words as keywords
• Also, provide a novelty statement at the end. What new things authors have done or correlated in this research compared to old ones?
• Would you please give a single line about the knowledge gap which your research has covered along with the hypothesis statement?
• If the authors are not sure, then give future recommendations for more research and investigation.
• Add the targeted beneficiary audience who will get benefits from this research.
• Also, give clear-cut recommendations and future prospective regarding this research.
This research shows the important scientific progress in content, some literature with related aspects should be cited for broader readership, such as

https://doi.org/10.1038/s41598-021-89103-9
https://doi.org/10.1016/j.jhazmat.2021.126493
https://doi.org/10.1016/j.envpol.2021.117854
https://doi.org/10.3390/ijerph17197251
https://doi.org/10.1038/s41598-021-01337-9

---

## Round 0.2 · accepted · Accept

Dear Authors,

I'm pleased to inform you that your manuscript have been accepted for publication

Reviewer 2 ·

Basic reporting

Is the review of broad and cross-disciplinary interest and within the scope of the journal?
Yes

Has the field been reviewed recently? If so, is there a good reason for this review (different point of view, accessible to a different audience, etc.)?
Yes

Does the Introduction adequately introduce the subject and make it clear who the audience is/what the motivation is?
Yes

Experimental design

Is the Survey Methodology consistent with a comprehensive, unbiased coverage of the subject? If not, what is missing?
Yes

Are sources adequately cited? Quoted or paraphrased as appropriate?
Yes
Article content is within the Aims and Scope of the journal.
Yes

Validity of the findings

Is there a well developed and supported argument that meets the goals set out in the Introduction?
Yes

Additional comments

The authors have addressed the comments and revised the manuscript successfully.